# NEUROlogical Prognosis After Cardiac Arrest in Kids (NEUROPACK) study: protocol for a prospective multicentre clinical prediction model derivation and validation study in children after cardiac arrest

Barnaby Robert Scholefield [1,2] James Martin,[3] Kate Penny-Thomas,[2] Sarah Evans,[2] Mirjam Kool,[1,2] Roger Parslow,[4] Richard Feltbower,[4] Elizabeth S Draper,[5] Victoria Hiley,[4] Alice J Sitch,[3,6] Hari Krishnan Kanthimathinathan,[1,2] Kevin P Morris,[2,3] Fang Smith,[1] NEUROPACK Investigators for the Paediatric Intensive Care Society-Study Group (PICS-SG)

For numbered affiliations see end of article.

**Correspondence to**
Dr Barnaby Robert Scholefield; b.scholefield@bham.ac.uk

## ABSTRACT

**Introduction** Currently, we are unable to accurately predict mortality or neurological morbidity following resuscitation after paediatric out of hospital (OHCA) or in-hospital (IHCA) cardiac arrest. A clinical prediction model may improve communication with parents and families and risk stratification of patients for appropriate postcardiac arrest care. This study aims to the derive and validate a clinical prediction model to predict, within 1 hour of admission to the paediatric intensive care unit (PICU), neurodevelopmental outcome at 3 months after paediatric cardiac arrest.

**Methods and analysis** A prospective study of children (age: >24 hours and <16 years), admitted to 1 of the 24 participating PICUs in the UK and Ireland, following an OHCA or IHCA. Patients are included if requiring more than 1 min of cardiopulmonary resuscitation and mechanical ventilation at PICU admission Children who had cardiac arrests in PICU or neonatal intensive care unit will be excluded. Candidate variables will be identified from data submitted to the Paediatric Intensive Care Audit Network registry. Primary outcome is neurodevelopmental status, assessed at 3 months by telephone interview using the Vineland Adaptive Behavioural Score II questionnaire. A clinical prediction model will be derived using logistic regression with model performance and accuracy assessment. External validation will be performed using the Therapeutic Hypothermia After Paediatric Cardiac Arrest trial dataset. We aim to identify 370 patients, with successful consent and follow-up of 150 patients. Patient inclusion started 1 January 2018 and inclusion will continue over 18 months.

**Ethics and dissemination** Ethical review of this protocol was completed by 27 September 2017 at the Wales Research Ethics Committee 5, 17/WA/0306. The results of this study will be published in peer-reviewed journals and presented in conferences.

**Trial registration number** NCT03574025.

### Strengths and limitations of this study

► This protocol has followed the international recommended Transparent Reporting of a multivariable prediction model for Individual Prognosis or Diagnosis guidelines for the derivation and validation of a clinical prediction model of neurodevelopmental outcome after paediatric cardiac arrest.
► A nationwide study which will efficiently combine routinely collected data through the existing, high-quality, Paediatric Intensive Care Audit Network database and a bespoke research database.
► Personalised recruitment and local follow-up will aim to maximise participant retention.
► The low incidence and wide variety of causes of paediatric cardiac arrest may restrict number of available patients and are potential limitations in prospective prognostic research in this population.
► Baseline neurodevelopmental status of patients will only be allocated retrospectively using the Paediatric Cerebral Performance Category tool.

## INTRODUCTION
### Paediatric cardiac arrest

Paediatric cardiac arrest (CA) is an uncommon but potentially catastrophic event for both children and their families. CA is defined as the cessation of cardiac mechanical activity occurring with absence of signs of circulation. Approximately 1500 infants or children per year suffer a CA in the UK and Ireland (RoI) with between 250 and 350 admitted to a paediatric intensive care unit (PICU) for postresuscitation care.[1] Survival to PICU discharge for this population is

achieved in 35%–45% patients admitted to PICU after an out of hospital CA (OHCA) and 45%–55% after in-hospital CA (IHCA). However, 50% of survivors are estimated to have ongoing neurodevelopmental disabilities despite advances in post-CA management.[2][3] The high mortality and morbidity rates are often associated with the degree of brain injury from the hypoxic-ischaemic insult at the time of CA.

## Prognostication after CA

Clinicians are currently unable to accurately predict survival with a good neurodevelopmental outcome after CA with any certainty due to a lack of data.[4–6] Clinicians can be pessimistic, optimistic or unnecessarily ambiguous in their predictions, and this affects the clarity of communication with families and the implementation of ongoing treatment plans.[4] Improved prognostication is, therefore, a high priority for parents of children who have suffered a CA. In addition, early stratification of patients who may benefit from critical care interventions would also be a significant advancement in their treatment[7][8] and has been lacking in major studies to date.[2][3]

Several prognostic factors are associated with survival following paediatric CA, such as patient age and pre-existing comorbidities,[9] CA characteristics (location, initial CA rhythm, duration of CA, presence and actions of bystanders,[9][10] physiological observations (eg, pupillary response, blood lactate, systolic blood pressure)[1][10][11] and specific medical interventions.[11][12] However, studies examining prognostic factors for good neurodevelopmental outcome are much less frequent.

The importance and weighting of these factors in prognosis decision making is complex and in 2010 the International Liaison Committee On Resuscitation (ILCOR) consensus statement identified a significant gap in knowledge in prognostic modelling with children[5] with no additional 'high-quality' data to inform the 2015 guidance.[13]

## Rationale for study

Accurate early prediction of neurodevelopmental outcomes may reduce uncertainty and improve communication with families. It may also provide better risk stratification for clinical trials and individualised treatment of patients. Furthermore, we aim to gain a better understanding of the epidemiology and neurodevelopmental outcomes of children after CA in the UK and RoI.

## METHODS AND ANALYSIS
## Study aims

The aim of the NEUROlogical Prognosis After Cardiac Arrest in Kids study is to (1) derive a clinical prediction model using key factors prospectively collected from a cohort of patients, available within the first hour of PICU admission after paediatric CA to predict good neurodevelopmental outcome at 3 months, (2) externally validate the clinical prediction model using an existing paediatric

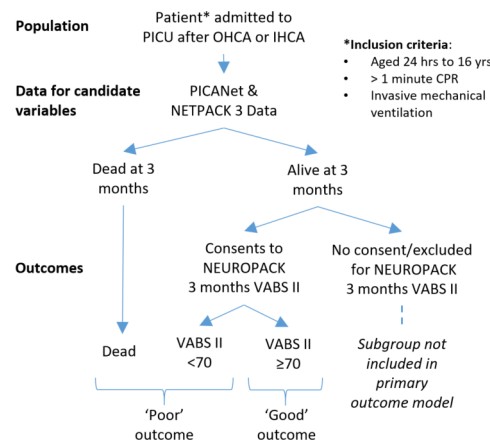

**Figure 1** NEUROPACK study overview: population, data collection tools and primary outcome. CPR, cardiopulmonary resuscitation; IHCA, in-hospital cardiac arrest; NEUROPACK, NEUROlogical Prognosis After Cardiac Arrest in Kids; OHCA, out of hospital cardiac arrest; PICU, paediatric intensive care unit; PICANET, Paediatric Intensive Care Audit Network; NETPACK 3, PICANet Post Arrest Care in Kids audit; VABS II, Vineland Adaptive Behavioural Score 2nd ed.

CA dataset and (3) describe the current epidemiology of CA cases in the UK and Ireland (RoI).

## Study design

This study is a multicentre, nationwide, prospective observational study combining both registry and cohort data. See figure 1 for study overview.

## Setting

Patients will be enrolled from 24 PICUs within the UK and RoI. All study sites admit infants and children following CA and routinely submit audit data to the Paediatric Intensive Care Audit Network (PICANet) registry.

## Ongoing PICU registry: PICANet and NET-PACK 3

Since 2002, PICANet has prospectively collected demographic, diagnostic, and interventional data along with PICU survival outcomes for patients admitted to PICUs in England and Wales and now collects data for patients across the UK and RoI.[14] This includes severity of illness variables to build the Paediatric Index of Mortality (PIM) risk-adjustment models.[15]

PICANet is also conducting an ongoing customised data collection of post-CA management: PICANet Post Arrest Care in Kids (NET-PACK 3) with data definition and data collection form (online supplemental materials 1 and 2). NET-PACK 3 customised data collection includes resuscitation variables available within a few hours of the CA. Data are either collected within 1 hour of admission onto PICU or within 1 hour of the attendance at the patient's bedside of a specialist paediatric critical care team (eg, a specialised retrieval team travels to another hospital without a PICU). These variables include: (1) attempted bystander cardiopulmonary resuscitation (CPR), (2) duration of CPR, (3) requirement of CPR after arrival at emergency department, (4) number

of doses of epinephrine (epinephrine) required and (5) initial presenting cardiac rhythm. These factors were chosen to comply with Utstein style CA reporting guidelines.[16 17] PICANet collects survival to PICU discharge outcome data for all admissions.

## ELIGIBILITY FOR NEUROLOGICAL PROGNOSIS AFTER CARDIAC ARREST IN KIDS
### Inclusion
All patients aged 24 hours up to 16th birthday admitted to PICU after OHCA or IHCA will be included. CA will be defined as requiring >1 min CPR. Patients will be included if they require invasive (eg, via endotracheal or tracheostomy) mechanical ventilation at PICU admission.

### Exclusion
Exclusion criteria include CAs occurring within a PICU or neonatal intensive care unit. For children who survive to PICU discharge we will exclude patients where the local clinical team at participating sites feel inclusion is inappropriate and/or parent/guardian or family member of children are unable to understand the telephone questionnaires for neurodevelopmental outcome assessments in English. All patients under the age of 24 hours will be excluded due to potentially different aetiology of CA related to birth events.

### Identification and screening
Patients for the NEUROlogical Prognosis After Cardiac Arrest in Kids (NEUROPACK) study will be identified via entry into the PICANet database and by local researchers at each site screening PICU admissions daily. 'CA preceding ICU admission—out of hospital or in-hospital' is a specific high risk category in the PIM-3 risk adjustment model and is recorded within 1 hour of PICU admission, or within 1 hour of the attendance at the patient's bedside of a specialist paediatric critical care team.[15]

### Recruitment for neurodevelopmental outcome assessment
Parent/guardians of CA patients who are expected to survive to 3 months following CA will be approached by local research staff, trained in Good Clinical Practice, to consent for telephone questionnaire at 3 months post-CA.

This is a very sensitive and difficult time for parents and guardians. The approach to parents or guardians of critical ill children for recruitment to the NEUROPACK study will, therefore, be handled sensitively. Local researchers will be trained to identify the appropriate time to consent, use passive information giving to reduce burden of information (eg, Ethics committee-approved posters displayed in family rooms) and liaise with the medical team managing the patient to acknowledge ongoing clinical management issues. Local site investigator (or delegate) will recontact parents or guardians at 2 months following CA to ascertain continued involvement in the study and to confirm ongoing contact details.

### Potential predictive factors collected
Potential candidate variables for the NEUROPACK clinical prediction model have been selected from the existing clinical prediction models for survival.[1 6 13] Final candidate variable selection will follow assessment of statistical modelling interaction and practicality of collecting variables in a timely fashion at the bedside by clinicians.

### Data collections
The ongoing NET-PACK 3 customised data collection and PICANet data collection for the PIM3 risk of mortality will be the data source for all the candidate variables in the NEUROPACK study. Linkage of individual patient NET-PACK 3 data with the collected neurodevelopmental outcome will be carried out for consented patients only. Pseudonymised data from NET-PACK 3 customised data collection and PICANET will be used for patients who die or for patients who survive and consent for follow-up assessment is not available.

## PRIMARY AND SECONDARY OUTCOMES
### Primary outcome
The primary outcome is survival with a good neurodevelopmental outcome at 3 months postevent. Good neurodevelopmental outcome is defined as a Vineland Adaptive Behaviour Scales second edition (VABS-II) score of ≥70.[18]

### Primary outcome assessment
The VABS-II was designed as a caregiver report measure to assess communication, daily living, social and motor domains of adaptive behaviour.[18] This tool can be used across the entire paediatric age range (0–16 years) and requires a short interview which can be via telephone. VABS-II is sensitive to neurological injury and has been used successfully in paediatric neurocritical care studies.[2] VABS-II has a normal mean value score of 100 (SD of 15). Good neurodevelopmental outcome is defined as a score of ≥70. Poor outcome is a composite score of VABS-II <70 and death. The chief investigator or the lead research nurse at the Central Research Centre (Birmingham Women and Children's National Health Service (NHS) Foundation Trust, UK) will conduct all assessments. At the time of outcome assessment, the assessor will remain blinded to the clinical prediction model and component variables.

### Secondary outcomes
Paediatric cerebral performance category (PCPC) and paediatric overall performance category (POPC) at 3 months and change in PCPC and POPC score from baseline.[19] Survival to PICU discharge and 3 months post-CA.

### Secondary outcome assessment
PCPC and POPC scale can be calculated by a short questionnaire conducted at the 3-month follow-up interview for consented patients. A baseline (pre-CA) PCPC and POPC will also retrospectively ascertained at the 3-month follow-up. PCPC and POPC have been recommended for

reporting in all paediatric CA studies. They score 1–6 (1: normal, 2: mild disability, 3: moderate disability, 4: severe disability, 5: vegetative state or coma and 6: death). They provide less detail but correlate reasonably well with VABS II.[20] This will allow comparison with other CA studies. Good neurodevelopmental outcome will be defined as PCPC score of 1–3 or no change from baseline. Poor outcome will be defined as a score of 4 or more, including death. Three months follow-up time point is chosen following the ILCOR, core outcome set for adults after CA recommendation[21] and demonstration of minimal change between three and 12-month following CA.[22]

## STATISTICAL CONSIDERATION
### Data analysis plan
The data will be manually reviewed for errors, missing data and outliers before analysis. Extreme values will be set to missing if they are deemed unlikely, based on their validity range. Descriptive analysis of the data will be reported. Continuous variables will be reported as either median and interquartile range (IQR) or mean and standard deviation (SD) based on the distribution. Categorical variables will be described in numbers, percentages or both.

### Sample size
To reduce problematic bias and improve precision we aim for at least 10 events per variable considered for multivariable modelling.[23] Following pilot data collection, we calculate 250 CA patients per year are admitted to 27 UK and RoI PICUs, 125 (50%) will survive to PICU discharge and 70 (30%) per year will survive with good neurodevelopmental outcome. To test seven variables we estimate a requirement of 70 events (eg, patients with good neurodevelopmental outcome). One hundred per cent of non-survivors will be included (included in PICANet and NET-PACK 3 audit database). We anticipate 80% recruitment and consent rate of remaining survivors. We, therefore, require data collection over an 18-month period to recruit 370 patients. We anticipate that this would ensure successful consent and follow-up of 150 patients, of whom 75 patients are estimated to have a good neurodevelopmental outcome.

### Statistical methods for developing a prognostic model
We will develop a prognostic model using logistic regression analysis of candidate variables and a good neurodevelopmental outcome as the primary outcome variable. Multiple imputation (using chained equations) will be used for any variables with missing data considered in the model. Auxiliary variables will be used to aid the imputation. The number of imputed data sets used will be equal to the fraction of missing data.[24]

Box 1 lists all candidate variables. Those variables deemed to be clinically important will be forced into the final model. Candidate variables will be retained if they benefit the model. The process will begin by fitting

> ### Box 1  Patient and cardiac arrest characteristics
>
> **Patient Demographic**
> - Age in years.*
> - Presence of PIM-3 'high-risk' comorbidities.†[15]
>
> **Cardiac arrest characteristics and interventions**
> - Location of cardiac arrest (IH & OHCA).†
>   OHCA is assigned if chest compressions were initiated before hospital arrival.
> - Aetiology of arrest (cardiac and non-cardiac).†
> - Duration of cardiopulmonary resuscitation.*
> - Continuation of cardiopulmonary resuscitation after Emergency Department arrival (for OHCA only).†.
> - Bystander cardiopulmonary resuscitation.†
> - Initial cardiac rhythm recorded during CA (shockable and non-shockable).†
> - Doses of epinephrine (epinephrine) during cardiopulmonary resuscitation.*
> - Use of continuous vasoactive infusions within 1 hour of PICU admission.†
>
> **Service characteristics**
> - Requirement of inter-hospital transfer prior to PICU admission.†
> - Time of arrest day (07:00–18:59) or night (19:00–06:59).†
>
> **Physiological variables**
> Measured for PIM-3 calculation: within 1 hour of PICU admission or within 1 hour of the attendance at the patient's bedside of a specialist paediatric critical care team
> - Systolic blood pressure.*
> - Pupillary reaction to light (greater than 3 mm and both fixed and other).†
> - Blood lactate level.*
>
> *continuous data, †categorical data.
> IH, in-hospital; OHCA, out-of-hospital cardiac arrest; PIM-3, Paediatric Index of Mortality 3 score.

the full model and then performing backwards elimination, with a conservative significance level of 0.157.[25] For categorical variables, the category with the lowest p value will dictate whether the variable is included in the final model.

All continuous variables will be left in their raw form to ensure no data were lost through dichotomisation or categorisation. It will be initially assumed that variables follow a linear trend, before fractional polynomials will be considered using the following powers: -2, –1, –0.5, natural logarithm, 0.5, 1, 2, and 3. A p<0.001 will be required to use a fractional polynomial rather than assuming a linear trend.[26] The use of fractional polynomials will also be considered for all continuous variables eliminated from the model to check whether this changes their inclusion status.

### Assessment of prognostic model performance
Assessment of the fitted model will be achieved by estimating calibration and discrimination. A calibration plot will be produced by plotting the observed risk against the predicted risk and the calibration slope calculated. We expect the slope should be approximately one as the model developed will be developed using this data.

To judge discrimination, the area under the receiver operating curve (equivalent to the C-statistic) and the R squared statistic will be calculated.

### Internal validation of the prognostic model

The model will be internally validated using bootstrap methods. The original data will be used to generate 100 bootstrapped data sets. Each one of these bootstrapped data sets will then be used to develop a prognostic model in the same way as the original model. Estimates of performance (C-statistic and calibration slope) will be obtained from the model fitted using each of the bootstrapped data sets. The estimates obtained from the bootstrapped data sets will be averaged and subtracted from the estimates from the original model to estimate optimism and provide optimism-adjusted performance statistics.

### Final prognostic model

The optimised adjusted calibration slope will then be used as a uniform shrinkage factor. Each of the coefficients from the original model will be adjusted for by multiplying by the shrinkage factor. The intercept will also be adjusted to ensure calibration-in-the-large, the average predicted probability, is the same as the average observed probability.

### Secondary analysis

Using the secondary outcomes, we will repeat the steps above to create a supplemental final prognostic model, for survival to PICU and survival to 3 months. In addition, we will create a prognostic model for good neurodevelopmental outcome using POPC and PCPC outcome scores.

There is a potential for survivors to decline consent, be lost to follow-up, or fulfil the exclusion criteria into the NEUROPACK study, and therefore, there is a risk that the survival subgroup may be biased. We plan to undertake sensitivity analyses by (1) imputing missing VABS II score for survivors using their known PICANet and NETPACK 3 data, (2) assume all survivors without a neurodevelopmental score had a VABS II score ≥70 and (3) assume all survivors without a neurodevelopmental score had a VABS II score <70, to ascertain impact of this group on the final prognostic model.

In addition, due to the limitations of not having a baseline VABS II score, we will also perform a secondary analysis using VABS II score ≥70 as the good neurodevelopmental outcome for a subgroup of patients with a known baseline PCPC score 1–3. This will allow comparison of the final prognostic model for all patients and the subgroup with known good neurodevelopment outcome at baseline.

### External validation of the NEUROPACK prognostic score

As part of the process of ensuring a prediction model is considered clinically useful, it must be validated in an external dataset.[27] We aim to do this by validating the NEUROPACK prognostic model in the publically accessible dataset for the Therapeutic Hypothermia After Paediatric Cardiac Arrest OHCA and IHCA randomised controlled trials in the National Institute for Health Biolincc repository (Http://biolincc.nhlbi.nih.gov).[2][3] The sample size of the dataset to be used for external validation should be sufficient to provide reliable and accurate results. To externally validate the model, predictions of risk for each patient in the external validation dataset are made, and performance statistics, such as the C-statistic, are calculated in the same manner as described earlier.

## PATIENT AND PUBLIC INVOLVEMENT

Given the sensitive and emotive nature of the NEUROPACK study, and the need for active parent and family engagement throughout, a patient advisory group, consisting of parents with experience of critical illness and death in children, and the Clinical Research Network: Children young person's advisory group (a sub group of the Generation R group aged 9–17 years) have been consulted in designing the protocol, the informational material to support the intervention, and to understand the burden of the intervention from the patient's perspective. At the end of the study, the patient advisory group will be consulted on findings and contribute to the dissemination plan.

## ETHICS AND DISSEMINATION

PICANet has ethical approval as a research database granted by the East Midlands, Derby Research Ethics Committee (ref 18/EM/0267) and NHS Health Research Authority Confidentiality Advisory Group approval (ref PIAG 4–07/(c)2002) to collect personally identifiable data without consent. The PICANet Clinical Advisory Group has approved pseudonymised sharing of PICANet audit data for the NEUROPACK study and Data Sharing Agreements will be established with the data controllers for the PICANet dataset prior to the release of de-identified PICANet and NET-PACK 3 data. Quality control of NET-PACK 3 customised data collection, data definitions and data collection is performed by the PICANet team.

Regional Ethics Committee (REC) permission has been obtained (Wales Research Ethics Committee 5, 17/WA/0306). This permits the ethical approach and consent of parents/guardians of eligible children who are likely to survive to 3 months following CA to enable telephone VABS-II assessment and identified data-linkage and sharing with PICANet and NET-PACK3 data.

We aim to publish the results in peer-reviewed journals and present at relevant national and international conferences.

### Author affiliations
[1]Birmingham Acute Care Research Group, University of Birmingham College of Medical and Dental Sciences, Birmingham, UK
[2]Paediatric Intensive Care Unit, Birmingham Women and Children's NHS Foundation Trust, Birmingham, UK
[3]Institute of Applied Health Research, University of Birmingham, Birmingham, UK
[4]Leeds Institute for Data Analytics, University of Leeds, Leeds, UK

[5]Health Sciences, University of Leicester College of Medicine Biological Sciences and Psychology, Leicester, UK

[6]NIHR Birmingham Biomedical Research Centre, University of Birmingham, Birmingham, UK

**Acknowledgements** We would like to thank all participating sites for their contribution to data collection. We thank the NEUROPACK Principle Investigators from the Paediatric Intensive Care Society Study Group (PICS-SG) listed below: Alder Hey Children's Hospital, Liverpool, UK. Dr Kent Thorburn; Birmingham Women and Children's Hospital NHS Foundation Trust, UK. Dr Barnaby Scholefield; Bristol Royal Hospital for Children, UK. Dr Rohit Saxena; Freeman Hospital, Newcastle, UK. Dr Yamuna Thiru; Glasgow Royal Hospital for Children, UK. Dr Richard Levin; Great North Children's Hospital, Newcastle, UK. Dr Rachel Agbecko; Great Ormond Street Hospital, London, UK. Dr Timothy Thiruchelvam & Dr Sophie Skellett; Imperial College Healthcare NHS Trust, London, UK. Dr David Inwald.; John Radcliff Hospital, Oxford, UK. Dr James Weitz; Kings College London NHS Foundation Trust, London, UK. Dr Akash Deep; Leeds Children's Hospital, UK. Dr Sian Cooper; Leicester Children's Hospital, UK. Dr Peter Barry; Nottingham Children's Hospital, UK. Dr Patrick Davies; Our Ladies Children's Hospital, Crumlin, Dublin, Republic of Ireland. Dr Cormac Breatnach.; Royal Brompton Hospital, London, UK. Dr Sandra Gala-Peralta; Royal Hospital for Sick Children, Edinburgh, UK. Dr Milly Lo.; Royal Manchester Children's Hospital, UK. Dr Rachael Barber; Royal Victoria Hospital, Belfast, UK. Dr Stewart Reid.; Sheffield Children's NHS Foundation Trust, UK. Dr Rum Thomas.; Southampton General Hospital, UK. Dr John Pappachan.; St George's Hospital, London, UK. Dr Buvana Dwarakanathan.; The Noah's Ark Children's Hospital for Wales, UK. Cardiff. Dr Siva Oruganti.; The Royal London Hospital, UK. Dr Kalai Sadasivam.; University Hospitals of North Midlands, Stoke on Trent, UK. Dr Mark Bebbington.

**Collaborators** NEUROPACK Investigators for the Paediatric Intensive Care Society-Study Group (PICS-SG); Kent Thorburn; Rohit Saxena; Yamuna Thiru; Richard Levin; Rachel Agbecko ;Timothy Thiruchelvam; Sophie Skellett; David Inwald; James Weitz; Akash Deep; Sian Cooper; Peter Barry; Patrick Davies; Cormac Breatnach; Sandra Gala-Peralta; Milly Lo; Rachael Barber; Stewart Reid; Rum Thomas; John Pappachan; Buvana Dwarakanathan; Siva Oruganti; Kalai Sadasivam; Mark Bebbington

**Contributors** BRS: initiated the collaborative project, designed the study, designed data collection tools, wrote the statistical analysis plan, drafted and revised the paper. He is guarantor. JM: wrote the statistical analysis plan, revised the paper. KP-T designed data collection tools, revised the paper. AJS: wrote the statistical analysis plan, revised the paper. RP, RF, ESD and VH: advised on PICANet data utility, monitor's data collection for PICANET and revised the paper. SE, MK, HKK, KPM and FS: designed the study, drafted and revised the paper.

**Funding** BRS is funded by a National Institute for Health Research (NIHR) Clinician Scientist Fellowship (NIHR-CS-2015-15-016) for this research project.

**Disclaimer** This publication presents independent research funded by the National Institute for Health Research (NIHR). The views expressed are those of the author(s) and not necessarily those of the NHS, the NIHR or the Department of Health and Social Care.

**Competing interests** None declared.

**Patient and public involvement** Patients and/or the public were involved in the design, or conduct, or reporting, or dissemination plans of this research. Refer to the Methods section for further details.

**Patient consent for publication** Not required.

**Provenance and peer review** Not commissioned; externally peer reviewed.

**ORCID iD**
Barnaby Robert Scholefield http://orcid.org/0000-0002-6198-4985

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
