## [Reviewer comments · BMJ Open]

ARTICLE DETAILS

TITLE (PROVISIONAL)	NEUROlogical Prognosis After Cardiac Arrest in Kids (NEUROPACK) study: protocol for a prospective multicentre clinical prediction model derivation and validation study in children after cardiac arrest.
AUTHORS	Scholefield, Barnaby; Martin, James; Penny-Thomas, Kate; Evans, Sarah; Kool, Mirjam; Parslow, Roger; Feltbower, Richard; Draper, Elizabeth; Hiley, Victoria; Sitch, Alice; Kanthimathinathan, Hari; Morris, Kevin; Smith, Fang

VERSION 1 – REVIEW

REVIEWER	Alexis Topjian The Children's Hospital of Philadelphia
REVIEW RETURNED	23-Mar-2020

GENERAL COMMENTS	This is a nicely written methods paper. I have two comments. The methods are slightly confusing because of the multiple acronyms, PICANET, NET-PACK 3 and NeuroPACK. I became confused as to what which dataset you need to be in to be eligible for the study and how NET-PACK 3 was part of the overall study. It took several reads for me to understand. Suggestions for clarity. 1. Perhaps a Figure out the datasets and which patients are included in which dataset and who will have follow up would be helpful. Do all patients in PICANET have ancillary NET-PACK 3 data collected? If they do not are they eligible for Neuro-PACK? The NEURO-PACK intervention independent from PICANet is the follow up at 3 months? Are the candidate variable from NeuroPACK all from PICA NET and Net PACK-3? The authors mention that patients will be consented for follow up if they are expected to live three months. Because consent is only being obtained in those who survive to discharge and all those who are dead are included there is a clear risk of bias by subgroup. This is just a challenge of doing this type of work, but it would be good for this bias/limitation to be included into the statistical section.
--

REVIEWER	Matthew Kirschen Children's Hospital of Philadelphia, USA
REVIEW RETURNED	31-Mar-2020

GENERAL COMMENTS	This is an excellent study aiming to improve prognostication after pediatric cardiac arrest. A few comments on the study methods for consideration. 1) It is unclear how the investigators will deal with patients who have baseline neurodevelopment disabilities with respect to their primary outcome of VABS. They have accounted for this with secondary outcomes in looking at change in PCPC compared to baseline, but this same approach is not detailed for the primary outcome. 2) The investigators should clarify whether subjects will be included if they have a tracheostomy and are mechanically ventilated at baseline. By definition they would meet criteria for mechanical ventilation on ICU admission if they had 90s CPR due to trach dislodgment, but were at neurologic baseline on ICU admission. 3) The authors state, "Parent/guardians of CA patients who are expected to survive to three months following CA will be approached by local research staff, trained in Good Clinical Practice, to consent for telephone questionnaire at three months post CA." It is unclear what the investigators mean when they write "expected to survive to 3 mo". Does this include kids who remain neurologically devastated and the parents opt for ongoing technological support vs withdrawal of care? 4) It is unclear why the authors are limiting their data collection to the first hour after ICU admission, although this is the information that goes into PIM calculation. Patients can be quite dynamic during the first several hours of post cardiac care and it may be beneficial in a subset of patients to look at physiological or therapeutic variables further into the ICU course, rather than just at presentation. Being able to prognosticate within the first hour post-arrest may be beneficial, but confidence in that prognosis will likely improve with more data over time. These data may be more beneficial for stratification for potential therapies in addition to prognostication. The authors should also consider retaining EEG and neuroimaging data for correlation with their initial physiologic data and outcome variables. 5) Lastly, I'm concerned about the ability of the PIM to differentiate between patients with mild to moderate neurocognitive deficits, primarily because pupillary reactivity carries such a disproportionately large amount of weight in the PIM calculation. There are many patients who will have unreactive pupils immediately post arrest due to a variety of factors that then may have a reasonably favorable outcome. It may be sufficient for a dichotomous good vs poor outcome though. It may be helpful to trend pupillary reactivity and abstract more details about the neurologic exam if feasible.
---

REVIEWER	Omar Khalid, MD Nationwide Children's Hospital The Ohio State University
REVIEW RETURNED	07-Jun-2020

GENERAL COMMENTS	Very interesting study that will add a valuable information about the prognosis after cardiac arrest. One minor correction on Line 50 "score 4 or less" should be "score 4 or more".
--

VERSION 1 – AUTHOR RESPONSE

Comment	Reviewer one: Dr Alexis Topjian	Response
1	The methods are slightly confusing because of the multiple acronyms, PICANET, NET-PACK 3 and NeuroPACK. I became confused as to what which dataset you need to be in to be eligible for the study and how NET-PACK 3 was part of the overall study. It took several reads for me to understand. Suggestions for clarity. 1. Perhaps a Figure out the datasets and which patients are included in which dataset and who will have follow up would be helpful. Do all patients in PICANET have ancillary NET-PACK 3 data collected? If they do not are they eligible for Neuro-PACK? The NEURO-PACK intervention independent from PICANet is the follow up at 3 months?	We have created and included a new Figure 1 (inserted at the bottom of this document) which is now an overview of the NEUROPACK study including the population inclusion/exclusion criteria, data collection stage (using PICANet data and NETPACK3 data collection tools) and then the enrolment into NEUROPACK/VABSII follow up assessment. We believe this is now much clearer in describing the stages of involvement and the multiple components of NEUROPACK study. We have also incorporated handling within the data analysis plan of the subgroup of patients who are alive at 3 months; however, are not enrolled into NEUROPACK study.
2	Are the candidate variable from NeuroPACK all from PICA NET and Net PACK-3?	This is correct, all candidate variables are contained within the PICANet main audit data or the additional NETPACK 3 audit data. We have amended the text to clarify this. 'The ongoing NET-PACK 3 customised data collection and PICANet data collection for the PIM3 risk of mortality will be the data source for all the candidate variables in the NEUROPACK study'
3	The authors mention that patients will be consented for follow up if they are expected to live three months. Because consent is only being obtained in those who survive to discharge and all those who are dead are included there is a clear risk of bias by subgroup. This is just a challenge of doing this type of work, but it would be good for	We agree that we will be able to include all patients who die after cardiac arrest; however, may have survivors who decline consent/enrolment and therefore will not be able to ascertain their VABS II score. We have added this into figure 1 and included a statement in the statistical analysis plan to undertake a sensitivity analysis on this group and

	this bias/limitation to be included into the statistical section.	the impact of their omission on the prognostic model findings. We agree that this is an inherent limitation of this type of research and will be reported accordingly after analysis. 'There is a potential for survivors to decline consent, be lost to follow up, or fulfil the exclusion criteria into the NEUROPACK study and therefore there is a risk that the survival subgroup is biased. We plan to undertake a sensitivity analysis by assuming all survivors without a neurodevelopmental score had a VABS score ≥ 70 and also rerun the analysis assuming the same group all had a score <70 to ascertain impact of this group on the final model.'
	REVIEWER TWO: Dr Matthew Kirschen	
4	It is unclear how the investigators will deal with patients who have baseline neurodevelopment disabilities with respect to their primary outcome of VABS. They have accounted for this with secondary outcomes in looking at change in PCPC compared to baseline, but this same approach is not detailed for the primary outcome.	Thank you for pointing out this omission. As we are unable to obtain accurate baseline VABS II in a time critical way (eg within 24 hours of admission) in our pragmatic study design we opted for local sites to record the simplified baseline PCPC score. We accept that this is a limitation in our study design. Our analysis plan aims to create the prediction model with the VABS score at 3 months irrespective of baseline VABS score. However, we will compare the results of the primary analysis with our secondary analysis where baseline PCPC score is taken into account in the assessment of eventual neuro-prognostic outcome. We will also be able to undertake a post-hoc analysis by including only patients who had a PCPC score of 1-3 at baseline. We have added the use of only the PCPC baseline score to the Strength and Limitations section. 'Baseline neurodevelopmental status of patients will only be allocated retrospectively using the Pediatric Cerebral Performance Category (PCPC) tool' We have also added the following to the secondary analysis plan:

		'Due to the limitations of not having a baseline VABS II score, we will also perform a secondary analysis using VABS II score of ≥ 70 as the good neurodevelopmental outcome for a subgroup of patients with a known baseline PCPC score 1-3. This will allow comparison of the final prognostic model for all patients and the subgroup with known good neurodevelopment outcome at baseline.'
5	The investigators should clarify whether subjects will be included if they have a tracheostomy and are mechanically ventilated at baseline. By definition they would meet criteria for mechanical ventilation on ICU admission if they had 90s CPR due to trach dislodgment, but were at neurologic baseline on ICU admission.	Yes these patients would fulfil inclusion criteria if they have a definitive airway (tracheostomy) and on mechanical ventilation at the time of PICU admission. We will be able to track the number of these patients through the PICANet data collection. Although in our experience, these patient (tracheostomy plus ventilation at home) would not require automatic PICU admission if they returned immediately to baseline. We would therefore expect their PICU admission to indicate a significant severity of the event and be a useful group to include in the prognostic model. We have amended the text: 'Patients will be included if they require invasive (e.g. endotracheal or tracheostomy) mechanical ventilation at PICU admission.'
6	The authors state, "Parent/guardians of CA patients who are expected to survive to three months following CA will be approached by local research staff, trained in Good Clinical Practice, to consent for telephone questionnaire at three months post CA." It is unclear what the investigators mean when they write "expected to survive to 3 mo". Does this include kids who remain neurologically devastated and the parents opt	The description of 'expected to survive to 3 months' in the protocol is to allow local sites to consider when and whether to approach families in the difficult days or weeks after PICU admission. This will allow us to identify and recruit patients early who may have a short stay in PICU, but also delay approach for those with uncertain outcomes where later limitation of therapy or withdrawal of life support may be considered days or weeks later. The reviewer is correct in identifying the scenario where families who decide to continue life sustaining therapy in patients with devastating injury will be included in our survivor group and can be approached upto the three month outcome stage.

	for ongoing technological support vs withdrawal of care?	This group we feel is really important, as these patients will not be part of any 'self-fulfilling prophecy' group, and a true estimation of early prognostic factors on their eventual outcome can be made.
7	It is unclear why the authors are limiting their data collection to the first hour after ICU admission, although this is the information that goes into PIM calculation. Patients can be quite dynamic during the first several hours of post cardiac care and it may be beneficial in a subset of patients to look at physiological or therapeutic variables further into the ICU course, rather than just at presentation. Being able to prognosticate within the first hour post-arrest may be beneficial, but confidence in that prognosis will likely improve with more data over time. These data may be more beneficial for stratification for potential therapies in addition to prognostication. The authors should also consider retaining EEG and neuroimaging data for correlation with their initial physiologic data and outcome variables.	Our primary objective is to create a clinical prediction model that would better inform clinicians at the time of admission to PICU so that clinical management decisions could be made, and communication with families improved. We agree that after admission with time and additional information that the level of uncertainty on prognosis will reduce and the addition of neurophysiological and neuroimaging information at 24-72 hours will provide a much higher level of information. However, by these time points the window of opportunity to choose neuroprotective therapies will have gone. The efficiency of our study design is to utilise the existing PICANet data collection process within 1 hour of PICU admission (or arrival of a critical care team), which will allow us to collect data available to clinicians at admission to PICU when these early decisions are made. However, the reviewer makes a very good suggestion, which we will consider of collecting, where feasible, additional neuro-prognostication data as part of a supplementary study to inform prediction models designed at a later stage of the patient's journey.
8	Lastly, I'm concerned about the ability of the PIM to differentiate between patients with mild to moderate neurocognitive deficits, primarily because pupillary reactivity carries such a disproportionately large amount of weight in the PIM calculation. There are many patients who will have unreactive pupils immediately post arrest due to a variety of factors that then may have a reasonably favorable outcome. It may be sufficient for a dichotomous good vs poor outcome though. It may be	We completely agree with this helpful observation. Patients in this cohort will have all clinical components of PIM-3 recorded as part of the 1st hour of critical care risk assessment; however, we plan to extract the individual components of the PIM-3 model as detailed in table 1 (eg pupillary reaction, systolic blood pressure, blood lactate level) and use these as separate independent variables within the prognostic model, rather than a single PIM-3 probability of death value. As described above, we are limited by the data collection design to only use clinical information

	helpful to trend pupillary reactivity and abstract more details about the neurologic exam if feasible.	from within the first hour of PICU admission/critical care management. We hope in future studies to be able to collect more longitudinal data over the course of PICU management.
	REVIEWER THREE: Dr Omar Khalid	
	One minor correction on Line 50 "score 4 or less" should be "score 4 or more".	Thank you. We have corrected this error.

Figure 1: NEUROPACK Study Overview: Population, data collection tools and primary outcome assessment

VERSION 2 – REVIEW

REVIEWER	Alexis Topjian Children's Hospital of Philadelphia, US
REVIEW RETURNED	01-Jul-2020

GENERAL COMMENTS	Thank you for clarifying my concerns. My last comment is that I believe the VABS II is no longer in print and that it has been replaced by the VABS-3. The authors should review and update accordingly.
--

REVIEWER	Matt Kirschen CHOP, USA
REVIEW RETURNED	01-Jul-2020

GENERAL COMMENTS	No further comments. Appreciate the explanations and revisions to the manuscript. Figure 1 is very helpful.
---